# Newly Developed Recombinant Antithrombin Protects the Endothelial Glycocalyx in an Endotoxin-Induced Rat Model of Sepsis

**DOI:** 10.3390/ijms22010176

**Published:** 2020-12-26

**Authors:** Toshiaki Iba, Jerrold H. Levy, Koichiro Aihara, Katsuhiko Kadota, Hiroshi Tanaka, Koichi Sato, Isao Nagaoka

**Affiliations:** 1Department of Emergency and Disaster Medicine, Juntendo University Graduate School of Medicine, Tokyo 113-8421, Japan; aihara@juntendo.ac.jp (K.A.); k-kadota@juntendo.ac.jp (K.K.); 2Department of Anesthesiology and Critical Care, Duke University School of Medicine, Durham, NC 27710, USA; jerrold.levy@duke.edu; 3Department of Emergency and Critical Care Medicine, Juntendo University Graduate School of Medicine Urayasu Hospital, Chiba 279-0021, Japan; htanaka@juntendo-urayasu.jp; 4Department of Surgery, Juntendo Shizuoka Hospital, Juntendo University Graduate School of Medicine, Tokyo 113-8421, Japan; kou-sato@chive.ocn.ne.jp; 5Department of Host Defense and Biochemical Research, Juntendo University Graduate School of Medicine, Tokyo 113-8421, Japan; nagaokai@juntendo.ac.jp

**Keywords:** antithrombin, endothelium, glycocalyx, syndecan-1, hyaluronan

## Abstract

(1) Background: The endothelial glycocalyx is a primary target during the early phase of sepsis. We previously reported a newly developed recombinant non-fucosylated antithrombin has protective effects in vitro. We further evaluated the effects of this recombinant antithrombin on the glycocalyx damage in an animal model of sepsis. (2) Methods: Following endotoxin injection, in Wistar rats, circulating levels of hyaluronan, syndecan-1 and other biomarkers were evaluated in low-dose or high-dose recombinant antithrombin-treated animals and a control group (*n* = 7 per group). Leukocyte adhesion and blood flow were evaluated with intravital microscopy. The glycocalyx was also examined using side-stream dark-field imaging. (3) Results: The activation of coagulation was inhibited by recombinant antithrombin, leukocyte adhesion was significantly decreased, and flow was better maintained in the high-dose group (both *p* < 0.05). Circulating levels of syndecan-1 (*p* < 0.01, high-dose group) and hyaluronan (*p* < 0.05, low-dose group; *p* < 0.01, high-dose group) were significantly reduced by recombinant antithrombin treatment. Increases in lactate and decreases in albumin levels were significantly attenuated in the high-dose group (*p* < 0.05, respectively). The glycocalyx thickness was reduced over time in control animals, but the derangement was attenuated and microvascular perfusion was better maintained in the high-dose group recombinant antithrombin group (*p* < 0.05). (4) Conclusions: Recombinant antithrombin maintained vascular integrity and the microcirculation by preserving the glycocalyx in this sepsis model, effects that were more prominent with high-dose therapy.

## 1. Introduction

Coagulation activation in sepsis is recognized as a natural event for host-defense that prevents pathogen dissemination and localizes infection [1,2]. However, the resulting inflammatory injury can cause a microcirculatory disturbance that leads to tissue ischemia, organ failure, and death [3]. Therefore, the goal of anticoagulant therapy in sepsis is to limit excessive coagulopathy while treating the underlying cause along with adjunctive therapy including antibiotics.

Antithrombin is one of the most important physiological anticoagulants. It provides host protection as an anticoagulant, but also through inhibiting acute inflammatory proteases such as thrombin and elastase [4,5,6]. Antithrombin is known to increase its anticoagulant activity by binding to heparan sulfate on the endothelial glycocalyx [7]. A unique feature of antithrombin is its capability to attenuate glycocalyx injury [8]. Antithrombin reportedly binds to heparan sulfate and contributes to glycocalyx stabilization [8,9]. Since antithrombin levels decrease dramatically in sepsis with coagulopathy [10], the supplementation of antithrombin for septic-associated disseminated intravascular coagulation (DIC) is a potential therapeutic approach [11]. Despite the lack of robust evidence, the antithrombin repletion in sepsis-associated DIC empirically administered in Japan is based on guidelines [12]. However, it has been suggested that the dose (intended to recover 80% of the activity) used in Japan is insufficient [13].

To overcome this problem, a recombinant non-fucosylated antithrombin was developed, and this novel antithrombin was named AT-γ. AT-γ is produced using the CHO DG44 cell line, which is deficient in α-1,6-fucosyltransferase (FUT8), and has four oligosaccharides (α-isoform) but lacks a core fucose, giving it a similar bioactivity and half-life to those of plasma-derived antithrombin [14,15]. In healthy volunteers, a dose of 72 IU/kg of AT-γ was shown to be bioequivalent to 60 IU/kg of plasma-derived antithrombin (data on file; Kyowa Hakko Kirin Co., Ltd., Tokyo, Japan). Furthermore, a multicenter open-label, randomized, active-controlled, phase 3 study has demonstrated an equivalent DIC resolution rate and mortality [16]. We previously reported the protective effects of AT-γ on cultured endothelial cells in vitro [17], and the effects of AT-γ on the glycocalyx and microcirculation were examined in an animal model in the present study. As a result, AT-γ administration facilitated the maintenance of the glycocalyx and tissue circulation and suppressed organ injury. We also confirmed that these effects were more prominent with high-dose AT-γ administration. In the present study, we investigated the protective effects of AT-γ on the glycocalyx in an animal model of sepsis.

## 2. Results

### 2.1. Changes in Leukocyte Adhesion and Blood Flow

One hour after lipopolysaccharide (LPS) administration, endothelial leukocyte adhesion had begun. As shown in Figure 1, left, the intact endothelium appeared as a smooth homogeneous monolayer of endothelial cells, while the glycocalyx appeared as a gap between the red blood cell (RBC) column and the endothelium. Following injection of LPS, multiple structural changes in the endothelium were observed including disruption, irregularities, and thickening as well as loss of the glycocalyx layer and an increase in leukocytes adhering to the endothelium. The number of leukocytes adhering to the endothelial layer was 17.6 ± 1.46/field at 3 h, increasing to 29.6 ± 1.81/field at 6 h in the control group. Leukocyte adhesion was suppressed in the high-dose AT-γ group, and the differences were significant at 3 and 6 h after LPS administration (*p* < 0.05, respectively) (Figure 2, left). Decreases in RBC velocity compared to the control group occurred due to leukocyte adhesion, accumulation, and microcirculatory plugging. These changes were suppressed in the high-dose AT-γ group at 6 h (*p* < 0.05) (Figure 2, right). Significant differences in leukocyte adhesion and RBC velocity were recognized between the high-dose and low-dose AT-γ groups at 3 and 6 h, respectively.

Bottom left: at 6 h, endothelial cells were thickened and the surface became irregular. The glycocalyx layer was lost and the blood cell contacted directly to endothelium. The leukocytes adherent to the endothelium increased and the blood flow decreased. Upper right: representative image of the microvessels in the intestinal wall. Quality checks were automatically performed using the GlycoCheck™ software. Invalid vascular segments are marked in yellow and were automatically discarded, while all the valid vascular segments (green lines) were further analyzed. The valid vascular segments were then visualized as the valid microvasculature (red lines).

### 2.2. Changes in Perfused Boundary Region (PBR) and Red Blood Cell (RBC) Filling Percentage

Figure 1 (right panel) shows representative vessel images obtained using side-stream dark-field microscopy. The accompanying software automatically discriminated valid vascular segments (green lines) from invalid segments (yellow lines). Then, the valid vascular segments were visualized as valid microvasculature (red lines). Figure 3 (left and right panels) showed the changes in the perfused boundary region (PBR) and the RBC filling percentage. The data are shown as the ratios relative to the normal values. The PBR began to increase at 1 h after LPS administration and reached 1.46 ± 0.09-fold of the normal value at 6 h. This increase was significantly suppressed in the high-dose AT-γ group (*p* < 0.05). Meanwhile, the RBC filling percentage began to decrease at 1 h after LPS administration, and the difference between the control group and the high-dose AT-γ group was also significant at 6 h (*p* < 0.05).

### 2.3. Laboratory Data

The laboratory data are summarized in Table 1. Antithrombin activity was significantly elevated by treatment with low-dose or high-dose AT-γ (*p* < 0.05, respectively) and recovered to within the normal range in the high-dose AT-γ group. The platelet count was decreased to 17 × 10^3^/mm^3^ in the control group, and this decrease was attenuated by treatment with both low-dose and high-dose AT-γ (*p* < 0.05, respectively). Coagulation abnormalities represented by a prolongation of the activated partial thromboplastin time (APTT) and a decrease in fibrinogen were attenuated by the treatment with high-dose AT-γ (*p* < 0.05, respectively). Organ damage markers such as ALT and BUN were elevated at 6 h after LPS administration, and treatment with AT-γ significantly reduced the elevation of BUN (*p* < 0.05, in both groups) but did not have a significant effect on the ALT level. A decrease in albumin and an elevation in lactate were recognized in the control group, and these changes were significantly attenuated in the high-dose AT-γ group (*p* < 0.05, respectively).

### 2.4. Hyaluronan and Syndecan-1 Measurements

Changes in syndecan-1 levels is shown in Figure 4 (left panel). In normal rats, the plasma levels were <5.0 ng/mL. The syndecan-1 levels increased over time and reached 42.81 ± 1.35 ng/mL at 6 h in the control group. The syndecan-1 levels were significantly lower in the high-dose group (28.07 ± 1.50 ng/mL, *p* = 0.001) at 6 h. The plasma level of hyaluronan in the normal rats was below 50.0 ng/mL, and it increased to 255.3 ± 15.5 ng/mL at 3 h after LPS administration and then decreased at 6 h. The plasma hyaluronan level was significantly lower in both the low-dose group (187.6 ± 10.1 ng/mL, *p* = 0.012) and the high-dose group (153.3 ± 15.9 ng/mL, *p* < 0.001) at 3 h (Figure 4, right).

## 3. Discussion

Endothelial glycocalyx is a critical aspect of the vascular endothelium that regulates vascular permeability, flow, cellular interactions, and anticoagulation [8,18,19,20]. However, following vascular injury due to causes that include hypoxia [21], trauma [22], and infection [23], the glycocalyx is easily disrupted, and major components including syndecans and glycosaminoglycans (hyaluronan) are released that provide potential biomarkers for evaluating injury [24]. In septic patients with acute lung injury, respiratory distress syndrome [25] or DIC [26], syndecan-1 and hyaluronan are elevated, and may serve as biomarkers to follow for evaluating therapies to improve vascular and endothelial dysfunction. 

Potential pharmacologic approaches to protect the glycocalyx include corticosteroids [27] and heparinoids [28], however, they have not been investigated in clinical settings of sepsis [29]. Plasma including albumin and antithrombin also are reported to protect the glycocalyx and reduce shedding in an animal model of ischemia and reperfusion [19,30,31]. We have also reported the ability of purified antithrombin to preserve vascular integrity [32], and similar results also observed using a novel recombinant non-fucosylated antithrombin namely AT-γ in vitro [17]. In the present study, we intended to confirm the effects of AT-γ in the animal model. 

Decreased glycocalyx thickness is reported to correlate with the severity of sepsis [33]. The combination of side-stream dark-field imaging and an automated computer calculation system enable the evaluation of the glycocalyx thickness and microperfusion. Theoretically, the deeper penetration of RBCs into the glycocalyx, as expressed by an increased PBR, should be associated with a reduced thickness of the glycocalyx. Thus, an increased PBR should represent a reduction in endothelial barrier properties for RBC accessibility (the reciprocal of glycocalyx thickness) [34]. The RBC filling percentage was monitored as a marker of microvascular perfusion. Lee et al. [4] reported a negative correlation between PBR and the RBC filling percentage. Similarly, the PBR increased over time after LPS administration, while the RBC filling percentage decreased over time in our study; both of these changes were significantly suppressed by treatment with high-dose AT-γ.

As described above, the glycocalyx is important in preserving vascular integrity but is easily damaged in sepsis. Following injury, glycocalyx fragments including syndecan-1 and hyaluronan, have been studied as vascular injury biomarkers. Recent clinical studies have also reported that these biomarkers can be used as diagnostic or potential prognostic indicators [35,36]. In the present study, the syndecan-1 level increased over time, whereas the hyaluronan level peaked at 3 h after LPS administration, and a beneficial effect of AT-γ was observed at 6 h for syndecan-1 and at 3 h for hyaluronan. Smart et al. [37] measured the syndecan-1 and hyaluronan levels in septic patients and reported that the syndecan-1 concentration increased, while the hyaluronan concentration peaked at an earlier timing. This phenomenon can be explained by the difference in the locations of these components. Syndecan-1 is a membrane-attached glycoprotein, while hyaluronan is mainly distributed on the surface of the glycocalyx [38]. Thus, hyaluronan is released beginning at an earlier stage of sepsis, and hyaluronan release is then followed by the shedding of syndecans.

The plasma albumin level has also been used to evaluate vascular integrity. Due to its amphoteric nature, albumin binds to the glycocalyx to decreases the vascular hydraulic conductivity [29]. The glycocalyx also has a negative charge that facilitates the repulsion of albumin. Overall, injury and destruction of the endothelial glycocalyx increases permeability to facilitate intravascular albumin loss. In our current study, albumin levels were better preserved in the high-dose group we believe to be a result of glycocalyx preservation.

### Limitations

The present study has several limitations. First, our sepsis model was LPS injection to the rat. Unfortunately, only the short-term observation can be possible with this model, and the long-term effect and the effect on mortality should also be examined in an additional experiment utilizing cecal ligation and puncture (CLP). However, a CLP model is not suitable for the intravital microscopic observation. Second, the detailed mechanism responsible for the action of AT-γ was not characterized in this study. Other than the direct protection of the glycocalyx, AT-γ may help to maintain the glycocalyx by improving anticoagulation. In the same manner, thrombin-mediated inflammatory reactions expressed via the activation of protease-activated receptors (PARs) might be involved [39]. Other mechanisms, such as the stimulation of prostacyclin production, should also be considered [40]. In addition, since this experiment does not have the formulation buffer of AT-γ group and, therefore, the buffer effect cannot be excluded. Although the observation period of our present study was limited, intravital microscopic observation or side-stream dark-field imaging are not suitable for long periods of observation.

## 4. Materials and Methods 

### 4.1. Lipopolysaccharide Administration and the Sepsis Model

All animal procedures were performed in accordance with the guidelines for animal experimentation of the Japanese Association for Laboratory Animal Science. Therefore, the Ethical Committee for Animal Experiments of Juntendo University waived the need for the approval of the institutional review board. Wistar rats (Tokyo Laboratory Animals Science Co., Tokyo, Japan) that were 10–12 weeks old were studied. Following intraperitoneal administration of sodium thiopental (100 mg/kg, Pentothal; Sigma Chemical Co., St. Louis, MO, USA), lipopolysaccharide (LPS, *E. coli* O55-B5; Difco Laboratories, Detroit, MI, USA) was diluted with 0.15 mL of sterile saline and a dose of 8.0 mg/kg of was administered intravenously (IV), followed by IV 250 IU/kg (low dose) or 500 IU/kg (high-dose) of AT-γ (Japan Blood Products Organization, Tokyo, Japan). Control animals (vehicle) group received IV LPS and saline.

### 4.2. Evaluation of the Microcirculation Using Intravital Microscopy

Following a median incision under anesthesia, the abdomen was opened to expose the mesentery, and immobilized on a warmed stand. Intravital microscopy (*n* = 7 in each group) was used to examine the mesenteric microcirculation using the Eclipse Pol™ microscopic system (Nikon Co., Tokyo, Japan) at 1, 3, and 6 h following LPS administration. A total of 6 successive fields were selected for each animal that was recorded at 30 frames/second for 5 min using a high-vision recording system (α7 III; SONY Co., Tokyo, Japan). Two independent blinded examiners were used to count adherent leukocytes in each field that was defined as adhered to a venule and stationary for 30 s. To analyze red blood cell (RBC) velocity, venules of 20–50 micrometers were localized, and a high-speed camera (Memrecam GX-1; Nac Image Technology Inc., Tokyo, Japan) recorded images at 1, 3 and 6 h after the LPS injection. Particle image velocimetry (Digimo Co., Tokyo, Japan) was used to determined RBC velocity, with baseline considered as 1.0, and decreases in the ratio determined.

### 4.3. Glycocalyx Evaluation Using Side-Stream Dark-Field Imaging

Glycocalyx thickness in intestinal microvessels was evaluated using side-stream dark-field imaging with the GlycoCheck™ System (Microvascular Health Solutions Inc., Salt Lake City, UT, USA) (*n* = 7 in each group). We have previously reported the measurement protocol [41]. The GlycoCheck™ System automatically records images and evaluates vascular segments at 10-micrometer intervals recording a total of 40 frames that include 300 major vascular segments. We selected 6 successive fields for every animal, moving the camera to a different location after ≥3000 vascular segments had been evaluated. Also, the perfused boundary region (PBR) and the RBC filling percentage were calculated.

### 4.4. Blood Sampling and Measurement

Six hours following LPS infusion, 7 animals per group for each timing were sacrificed in an ether chamber. Blood samples, obtained from the inferior vena cava, were collected in citrate, centrifuged, and stored at −80 °C until assayed. Three additional no treated rats were used as controls for blood sampling. Laboratory testing included platelet count, APTT, fibrinogen, antithrombin activity, alanine aminotransferase (ALT), blood urea nitrogen (BUN), and albumin levels using standard analytic techniques. A blood gas analyzer (ABL715; Radiometer, Københaven, Denmark) was used to determine lactate levels. An enzyme-linked immunosorbent assay (ELISA) kit was used to measure syndecan-1 (Cloud-Clone Corp., Katy, TX, USA), and a hyaluronan Quantikine™ ELISA kit (R&D Systems Inc., Minneapolis, MN, USA) according to the manufacturer’s instructions.

### 4.5. Statistical Analysis

Data were analyzed using a one-way analysis of variance (ANOVA) with the Dunnett post-hoc test using statistical software (StatView II^TM^, Abacus Concepts, Piscataway, NJ, USA). Data were presented as the mean ± standard error (SE). Differences were considered statistically significant at *p* < 0.05.

## 5. Conclusions

AT-γ attenuated damage to the glycocalyx in a rat model of sepsis. The shedding of syndecan-1 and hyaluronan from the vascular endothelial surface was suppressed by AT-γ, and these effects were more effective in animals treated with a high dose of AT-γ. The observed effects were thought to have led to the preservation of the glycocalyx, maintenance of the microcirculation, and the prevention of subsequent organ dysfunction. Thrombin is thought to be the key mediator in sepsis [42], and the observed effects were considered to be expressed through inhibition of the coagulation cascades since APTT and fibrinogen levels were better maintained in the high-dose group. However, as the direct evidence of thrombin inhibition was not proven in this study, it should be further confirmed. Finally, the present experiment examined only the short-term effect of AT-γ and we cannot exclude that the effects are transient and thrombin inhibition does not have a significant impact on disease evolution.

## Figures and Tables

**Figure 1 ijms-22-00176-f001:**
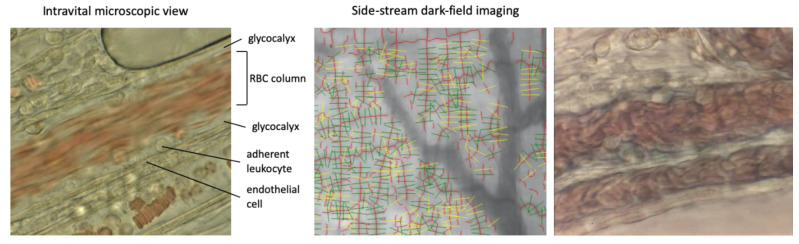
View of microvessels using intravital microscopy and image acquisition using the GlycoCheck™ System. Upper left: At baseline, the endothelium is visible as a smooth, thin layer, and the red blood cells (RBCs) are moving fluently. Some leukocytes have become stuck on the endothelium, but most of them are rolling. The glycocalyx layer is shown as a gap between the red cell column and the endothelial surface. 40× objective lens

**Figure 2 ijms-22-00176-f002:**
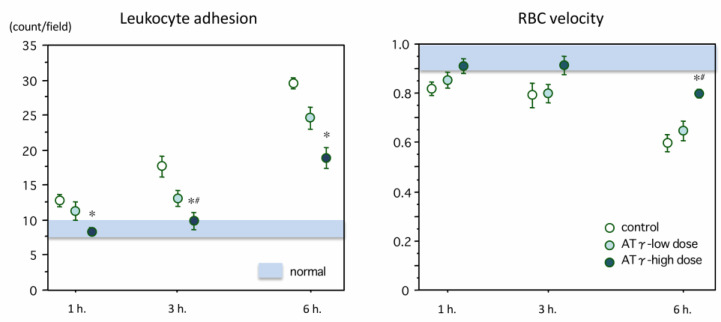
Changes in leukocyte adhesion and RBC velocity. Left: The adhesion of leukocytes onto the mesenteric vascular endothelium increased over time after lipopolysaccharide (LPS) administration in the control group, and the incidence of adherent leukocytes was significantly reduced in the high-dose AT-γ (novel antithrombin) group at 1, 3 and 6 h (*n* = 7 in each group). Right: The red blood cell (RBC) velocity in the mesenteric venule gradually decreased until 6 h after lipopolysaccharide administration. The velocity was significantly better maintained in the high-dose AT-γ group at 6 h. * *p* < 0.05, compared with the control group, ^#^
*p* < 0.05, compared with the low-dose group.

**Figure 3 ijms-22-00176-f003:**
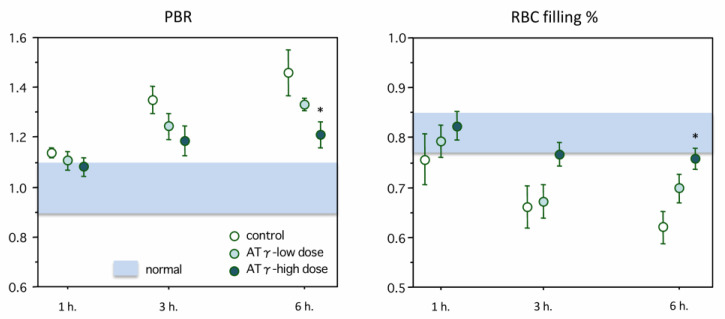
Changes in the perfused boundary region (PBR) and RBC filling percentage. Left: Table 1. 4-fold of the baseline value at 6 h after lipopolysaccharide administration. The increase in the PBR was significantly suppressed in the high-dose AT-γ group at 6 h (*n* = 7 in each group). Right: The changes in the red blood cell (RBC) filling percentage in the mesenteric microvessels are shown. The RBC filling percentage started to decrease after lipopolysaccharide administration and reached 62% of the baseline value at 6 h. The RBC filling percentage was significantly maintained better in the high-dose AT-γ group at 6 h. * *p* < 0.05, compared with the control group.

**Figure 4 ijms-22-00176-f004:**
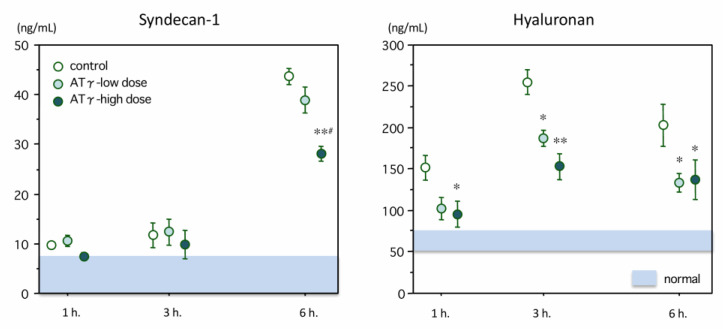
Changes in plasma levels of syndecan-1 and hyaluronan. Left: The syndecan-1 level was elevated at 6 h after lipopolysaccharide administration. The syndecan-1 level in the high-dose AT-γ group was significantly lower than that in the control group (*n* = 7 in each group). Right: The plasma level of hyaluronan peaked at 3 h after lipopolysaccharide administration. The hyaluronan levels were significantly suppressed in the AT-γ groups. * *p* < 0.05 compared with the control group, ** *p* < 0.01 compared with the control group, # *p* < 0.05 compared with the low-dose group.

**Table 1 ijms-22-00176-t001:** Laboratory findings of the endotoxemic rats treated with or without antithrombin.

Parameters	Platelet count (×10^3^/mm^3^)	*APTT* (second)	Fibrinogen (mg/dL)	*AT* activity (%)	*ALT* (IU/L)	*BUN* (mg/dL)	Albumin (mg/dL)	Lactate (mmol/L)
Normal group	110–120	16.1–16.7	210–220	100–120	15–32	12.7–14.5	4.7–5.1	0.8–0.9
Control group	17 ± 2	33 ± 3	82 ± 10	41 ± 10	144 ± 21	52 ± 3	2.3 ± 0.1	4.6 ± 0.3
Low-dose group	27 ± 3 *	31 ± 3	109 ± 11	82 ± 8 *	100 ± 13	35 ± 3 *	2.8 ± 0.1	3.6 ± 0.3
High-dose group	27 ± 3 *	24 ± 3 *	122 ± 12 *	101 ± 11 *	93 ± 16	37 ± 4 *	3.1 ± 0.2 *	3.4 ± 0.3 *

*APTT* activated partial thromboplastin time, *AT* antithrombin, *ALT* alanine aminotransferase, *BUN* blood urea nitrogen; * significant difference from the control group at *p* < 0.05.

## Data Availability

Data sharing not applicable

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
