# Peer review of "Newly Developed Recombinant Antithrombin Protects the Endothelial Glycocalyx in an Endotoxin-Induced Rat Model of Sepsis"

_ijms, 2020, doi:10.3390/ijms22010176_

Round 1
Reviewer 1 Report
In this study, Iba and colleagues showed that the blocking thrombin attenuates glycocalyx damage in an endotoxin-induced animal model of sepsis. Increases in systemic concentration of hyaluronan and syndecan-1 was reduced by AT-γ. The authors showed that treatment with and high dose of a recombinant protein AT-γ, soon after septic insult, was associated with lower leukocyte adhesion and better maintained blood flow. These data was supported by laboratory data: platelet-count, APTT, fibrinogen, BUN, albumin and lactate.
The paper is interesting and may contribute to define the still debated role of thrombin in sepsis.
However, I have some concerns:
Major concerns:
The use of the LPS-model to induce sepsis has some limitations; why did you prefer this model to the polymicrobial sepsis induced by CLP? From the scientific literature it emerges that the gold standard model of experimental sepsis is represented by CLP. It would be appropriate to confirm your findings using this model.
In this paper only a short-term analysis is reported. The authors did not evaluate the effect of Thrombin inhibition on survival and recovery after sepsis. Then, in the conclusions, they speculated that AT-γ treatment have led to a prevention of organ dysfunction, but they did not analyze organ protection. Thus, we cannot exclude that the effects of AT-γ are transient and/or that Thrombin inhibition has not a significant impact on disease evolution. This should be clearly stated and discussed in the paper.
Minor remarks:
In the section 2.3 Laboratory data the authors argue that the decrease or increase in some parameters, occurred during sepsis, is suppressed by treatment with high doses of AT; but the effect of this treatment is only partial and results in a reduction of the phenomenon, not with its suppression. I encourage the authors to use terms such as "reduce" or “attenuate”.
Figure 3 legend: the authors indicated that “PBR ……..reached 1.7-fold of the baseline value at 6 hours” but in the graph the mean value of PBR at 6 hours didn’t reach 1.5 fold of the baseline. Please correct.
Figure 4 legend: the p-value related to the double asterisk is missing.
Author Response
We would like to thank the reviewer for thoughtful comments. We have revised the manuscript as suggested and would like to address each comment as follows:
Reviewer #1
Major concerns:
The use of the LPS-model to induce sepsis has some limitations; why did you prefer this model to the polymicrobial sepsis induced by CLP? From the scientific literature, it emerges that the gold standard model of experimental sepsis is represented by CLP. It would be appropriate to confirm your findings using this model.
Reply
We appreciate your comment. We agree, the cecal ligation and puncture (CLP) polymicrobial sepsis model is increasingly used, but the purpose of our study was to observe the microcirculatory change of acute infection, and for consistency, we used an LPS model. In addition, we have previously studied the effects of plasma-derived antithrombin in the same endotoxin-induced rat model (Thromb Res.2018;171:1-6.) using the LPS model. To address your comment, we have now added to our report the potential limitations of our model and mention the need to confirm the effect in a CLP model (Page 7, Line 205-207).
In this paper, only a short-term analysis is reported. The authors did not evaluate the effect of Thrombin inhibition on survival and recovery after sepsis. Then, in the conclusions, they speculated that AT-γ treatment has led to the prevention of organ dysfunction, but they did not analyze organ protection. Thus, we cannot exclude that the effects of AT-γ are transient and/or that thrombin inhibition has not a significant impact on disease evolution. This should be clearly stated and discussed in the paper.
Reply
We appreciate your comments. As you indicated, the long-term effect and the effect on mortality should be examined in the next step. Since we mainly focused on the mechanism of better maintenance of microcirculation, i.e., endothelial glycocalyx protection, we used the LPS injection model in this experiment. As for organ protection, we showed lower levels of BUN in low- and high-dose AT-γ groups after 6 hr (Table 1). We added that thrombin inhibition is the key point, but state it was not shown directly in the present study in conclusion (Page 9, Line 269-272).
Minor remarks:
In section 2.3 Laboratory data, the authors argue that the decrease or increase in some parameters, occurred during sepsis, is suppressed by treatment with high doses of AT; but the effect of this treatment is only partial and results in a reduction of the phenomenon, not with its suppression. I encourage the authors to use terms such as "reduce" or “attenuate”.
Reply
Thank you for your advice. The terms were corrected accordingly.
Figure 3 legend: the authors indicated that “PBR ……..reached 1.7-fold of the baseline value at 6 hours” but in the graph, the mean value of PBR at 6 hours didn’t reach 1.5 fold of the baseline. Please correct.
Reply
We corrected the legend.
Figure 4 legend: the p-value related to the double asterisk is missing.
Reply
We apologize for this mistake. We corrected it in the legend.
Reviewer 2 Report
In the present work, Iba et al tested a recombinant non-fucosylated antithrombin (AT-g) as a protector of endothelial glycocalyx in an LPS induced model of sepsis is rats. Using intra-vital microscopy, laboratory explorations and measurement of syndecan-1 and Hyaluronan, they observed less leucocyte rolling, less coagulation activation and glycocalyx protection after high dose of AT administration.
Major Comments:
Even though this is a well conduct work, a major limitation is the absence of a control group with the formulation buffer of AT-g. Indeed, it was shown that some formulation buffers are not neutral, in particular those containing glycine (but not only), and at the doses infused. Thus, a buffer effect cannot be excluded. At least this point should have been discussed and indicated in the limitations of the study.
More generally, the introduction could more developed with more details on AT-g. In particular part of the discussion on non-fucosylated AT and its advantage compared to others AT should be moved to the introduction. The discussion should also be more diverse. Indeed, 20% of the cited references are from the authors. Although they have a good publication activity on this topic, more diversity and discussion about conflicting results is necessary.
A last issue is the absence of a control group (sham - PBS infection instead of LPS). It would be interesting to add these results if available, even in supplementary results.
Minor comments:
- Did the animals receive AT just after the end of LPS infusion?
- In Figure 1, it would be interesting to have a representative view of microvessels after LPS exposure to illustrate the description in the text (lines 67-70)
- In fig 2 and in figure 3, we have no indication about normal rats. It should be added, either in the text or in the figure legend
Author Response
We would like to thank the reviewer for thoughtful comments. We have revised the manuscript as suggested and would like to address each comment as follows:
Reviewer 2
Major Comments:
Even though this is a well conduct work, a major limitation is the absence of a control group with the formulation buffer of AT-g. Indeed, it was shown that some formulation buffers are not neutral, in particular those containing glycine (but not only), and at the doses infused. Thus, a buffer effect cannot be excluded. At least this point should have been discussed and indicated in the limitations of the study.
Reply
We appreciate your important comment. We now add “the experiment does not have the formulation buffer of AT-γ group and therefore, the buffer effect cannot be excluded” in the limitation (Page 7, Line 212-213).
More generally, the introduction could more developed with more details on AT-g. In particular part of the discussion on non-fucosylated AT and its advantage compared to others AT should be moved to the introduction. The discussion should also be more diverse. Indeed, 20% of the cited references are from the authors. Although they have a good publication activity on this topic, more diversity and discussion about conflicting results is necessary.
Reply
We appreciate your advice. The first 3 paragraphs in the discussion section were moved to the introduction section and a paragraph of introduction was moved to discussion accordingly. In addition, based on your points, the discussion section was rewritten. However, please understand there is a paucity of relevant studies that focus on this perspective.
A last issue is the absence of a control group (sham - PBS infection instead of LPS). It would be interesting to add these results if available, even in supplementary results.
Reply
We appreciate your important advice. The control was given physiological saline, and we thought the same amount of albumin could be another choice. However, since albumin also has a protective effect of glycocalyx (Ann Intensive Care. 2020, 10, 85.), it may not be appropriate as a control. In the previous study, we examined the effects of plasma-derived antithrombin in a similar model, and the saline group was set as the control (Thromb Res. 2018, 171, 1-6.), we set the saline group as a control.
Minor comments:
Did the animals receive AT just after the end of LPS infusion?
Reply
Yes, AT was infused just after LPS.
In Figure 1, it would be interesting to have a representative view of microvessels after LPS exposure to illustrate the description in the text (lines 67-70)
Reply
We added a figure and its legend accordingly.
In fig 2 and in figure 3, we have no indication about normal rats. It should be added, either in the text or in the figure legend
Reply
Thank you for your suggestion. We added the normal value in the figures accordingly.
The authors would like to thank the reviewer and the Editor for their comments and suggestions. Please let us know if any further changes are needed.
Sincerely,
Toshiaki Iba, MD
Jerrold H Levy, MD
Koichiro Aihara, MD
Katsuhiko Kadota, MD
Hiroshi Tanaka, MD
Koichi Sato, MD
Isao Nagaoka, MD
Round 2
Reviewer 1 Report
I have carefully evaluated the authors' responses to my observations and unfortunately they have not satisfied my major remarks despite the time since the first review.
Author Response
Dear Editor,
We would like to thank the reviewer for the comments. We further revised the manuscript and addressed the comments as follows:
Reviewer #1
Major concerns:
The use of the LPS-model to induce sepsis has some limitations; why did you prefer this model to the polymicrobial sepsis induced by CLP? From the scientific literature, it emerges that the gold standard model of experimental sepsis is represented by CLP. It would be appropriate to confirm your findings using this model.
Reply
As was stated in the previous response, we agree the cecal ligation and puncture (CLP) polymicrobial sepsis model is more adequate as a sepsis model, To address your comment, we have now changed the description to “Unfortunately, only the short-term observation can be possible with this model, and the long-term effect and the effect on mortality should also be examined in an additional experiment utilized cecal ligation and puncture. (Page 7, Line 205-207). We also added, “However, a CLP model is not suitable for the intravital microscopic observation” in the text.
In this paper, only a short-term analysis is reported. The authors did not evaluate the effect of Thrombin inhibition on survival and recovery after sepsis. Then, in the conclusions, they speculated that AT-γ treatment has led to the prevention of organ dysfunction, but they did not analyze organ protection. Thus, we cannot exclude that the effects of AT-γ are transient and/or that thrombin inhibition has not a significant impact on disease evolution. This should be clearly stated and discussed in the paper.
Reply
We further rewrote the Conclusion and added “Finally, the present experiment examined only the short-term effect of AT-γ and we cannot exclude that the effects are transient and thrombin inhibition does not have a significant impact on disease evolution.” (Page 9, Line 272-274).
The authors would like to thank the reviewer and the Editor for their comments and suggestions. Please let us know if any further changes are needed.
Sincerely,
Toshiaki Iba, MD
Jerrold H Levy, MD
Koichiro Aihara, MD
Katsuhiko Kadota, MD
Hiroshi Tanaka, MD
Koichi Sato, MD
Isao Nagaoka, MD
Reviewer 2 Report
I’m OK for the publication of this new version of the manuscriptAuthor Response
Thank you for your approval.
Round 3
Reviewer 1 Report
Iba et al. indicate that the purpose of their study is to evaluate the protective effect of the recombinant non-focusylated antithrombin (AT-γ) on the microcirculatory alterations during sepsis (LPS model). In a previous paper of this research group the same effect was demonstrated using plasma antithrombin (Thrombosis Research 2018).
The Authors also demonstrated that there are no differences between plasmatic antithrombin and AT-γ effects in vitro; therefore, the originality of this manuscript is limited to the confirmation in vivo of the AT-γ action already explored.
The suggestions given in the first review aimed:
- to increase the interest of the results by using a polymicrobial model more similar to human sepsis as reported by most of the literature (Doi K, J Clin Invest 2009).
- to demonstrate, with direct evidence, that AT-γ improves organ damage.
The Authors felt that the observations could be resolved by declaring the limits of the paper. In my opinion, the additional recommended experiments may amplify the interest of the manuscript through the study of the microcirculation in intravital microscopy that, although with greater difficulty, is also possible in the CLP model or by demonstrating the protective effect of AT-γ in organ damage. Therefore, in the absence of an additional experimental set, my previously sent opinion is confirmed.